# Altered Plasma microRNA Signature in Hospitalized COVID-19 Patients Requiring Oxygen Support

**DOI:** 10.3390/microorganisms12030440

**Published:** 2024-02-21

**Authors:** Sandra Franco, Lourdes Mateu, Raquel Pluvinet, Jose Francisco Sanchez-Herrero, Ruth Toledo, Lauro Sumoy, Marta Massanella, Miguel Angel Martinez

**Affiliations:** 1IrsiCaixa, Hospital Universitari Germans Trias i Pujol, 08916 Badalona, Spain; sfranco@irsicaixa.es (S.F.); mmassanella@irsicaixa.es (M.M.); 2Infectious Disease Service, Hospital Universitari Germans Trias i Pujol, 08916 Badalona, Spain; lmateu.germanstrias@gencat.cat; 3Institut Germans Trias i Pujol (IGTP), 08916 Badalona, Spain; rpluvinet@igtp.cat (R.P.); jsanchez@igtp.cat (J.F.S.-H.);; 4Fundació Lluita contra les Infeccions, Hospital Universitari Germans Trias i Pujol, 08916 Badalona, Spain; 5Centro de Investigación Biomédica en Red de Enfermedades Infecciosas (CIBERINFEC), 28029 Madrid, Spain

**Keywords:** SARS-CoV-2, miRNA, biomarker, COVID-19 prognosis, disease severity

## Abstract

To discover potential micro(mi)RNA biomarkers of SARS-CoV-2 infection and disease progression, large-scale deep-sequencing analysis of small RNA expression was performed on plasma samples from 40 patients hospitalized for SARS-CoV-2 infection (median 13.50 [IQR 9–24] days since symptoms initiation) and 21 healthy noninfected individuals. A total of 1218 different miRNAs were identified. When compared with healthy noninfected donors, SARS-CoV-2-infected patients showed significantly (fold change [FC] > 1.2 and adjusted p [padj] < 0.05) altered expression of 190 miRNAs. The top-10 differentially expressed (DE) miRNAs were miR-122-5p, let-7b-5p, miR-146a-5p, miR-342-3p, miR-146b-5p, miR-629-5p, miR-24-3p, miR-12136, let-7a-5p, and miR-191-5p, which displayed FC and padj values ranging from 153 to 5 and 2.51 × 10^−32^ to 2.21 × 10^−21^, respectively, which unequivocally diagnosed SARS-CoV-2 infection. No differences in blood cell counts and biochemical plasma parameters, including interleukin 6, ferritin, and D-dimer, were observed between COVID-19 patients on high-flow oxygen therapy, low-flow oxygen therapy, or not requiring oxygen therapy. Notably, 31 significantly deregulated miRNAs were found, when patients on high- and low-flow oxygen therapy were compared. SARS-CoV-2 infection generates a specific miRNA signature in hospitalized patients. Specific miRNA profiles are associated with COVID-19 prognosis in patients requiring oxygen flow.

## 1. Introduction

The clinical characteristics of coronavirus disease 2019 (COVID-19) range from asymptomatic signs to severe or critical conditions [1]. Intriguingly, a significant percentage of patients develop a long-lasting symptomatology after acute infection [2]. The spread of new severe acute respiratory syndrome coronavirus 2 (SARS-CoV-2) variants has also impacted COVID-19 pathogenesis and symptoms [3]. Nevertheless, the mechanisms underlying the different COVID-19 courses remain unidentified.

MicroRNAs (miRNAs) are 19–22 nucleotide noncoding RNAs that act as negative regulators of translation. They are involved in numerous cellular processes [4] and are key regulators of biological processes in animals. The deregulation of gene expression by miRNAs plays a major role in many human diseases, including cancer, cardiovascular disease, diabetes, and neurodegenerative disease [5]. miRNAs are secreted into extracellular fluids and delivered to other cells, in which they can also function [6]. Cell and tissue-specific miRNA expression and circulating miRNA signatures are likely to be valuable diagnostic tools for specific diseases. We and others have previously described how circulating miRNA profiles can be a useful tool for human virus infection diagnosis and prognosis [7,8].

Prior studies have identified a modified miRNA expression profile in COVID-19 patients [9,10]. The implication of miRNAs in the pathogenesis of COVID-19 has also been explored recently [11]. Moreover, it has been suggested that miRNAs may be novel therapeutic targets for the treatment of COVID-19 [12]. However, these studies have not been conclusive regarding changes in miRNA expression after SARS-CoV-2 infection and the underlying mechanisms associated with SARS-CoV-2 infection and COVID-19 development. The possible role of miRNAs in COVID-19 pathogenesis and disease course also remains unresolved. Here, we performed high-throughput sequencing of the plasma miRNAs from 40 acute hospitalized SARS-CoV-2-infected patients and 21 healthy noninfected individuals. In addition, we stratified SARS-CoV-2 patients between those requiring high- and low-flow oxygen therapy and no oxygen support at the time of hospitalization. We aimed to identify potential miRNA biomarkers of SARS-CoV-2 infection and disease progression.

## 2. Materials and Methods

### 2.1. Study Cohort

This cross-sectional study included plasma samples from 40 hospitalized patients with acute SARS-CoV-2 from the KING observational study at the Hospital Universitari Germans Trias i Pujol (Badalona, Spain), which aimed to characterize virological and immunological features of SARS-CoV-2 infection [13]. Participants were enrolled after a positive test for SARS-CoV-2 infection (either a virological test performed by RT–qPCR analysis of nasopharyngeal swabs in routine clinical screenings, or a serological test performed by in-house ELISA of plasma samples). Patient plasma samples were taken at the time of hospitalization. Plasma samples from a previously described [14] cohort of 21 healthy, uninfected volunteers were also analyzed as a control group. This study was approved by the Hospital Universitari Germans Trias i Pujol Ethics Committee Board (reference PI-20-122).

### 2.2. RNA Extraction

RNA was extracted with a miRCURY Isolation kit Biofluids (Exiqon, Vedbaek, Denmark), as we previously described [15]. Tests of the robustness of the plasma RNA isolation procedure and the detection of hemolysis were performed as previously described [15]. Plasma RNA was reverse transcribed, and a real-time PCR (RT-PCR) amplified with LNA-enhanced primers in the ExiLENT SYBR Green master mix (Exiqon) on a Lightcycler 480 RT-PCR system (LC480, Roche, Basel, Switzerland) was conducted. RT quantitative PCR (RTqPCR) Cp values were determined with the second derivative method provided in the LC480 software 1.5.

### 2.3. Small RNA Library Generation and Sequencing

Small RNA sequencing libraries were prepared according to the TruSeq Small RNA Sample Preparation Guide (Illumina, San Diego, CA, USA), as we previously described [15]. Prior to pooling and size selection, individual library yields and molarities were determined, based on an integration of the 135–160-bp size range, with the DNA 1000 Kit on a Bioanalyzer 2100 (Agilent Technologies, Santa Clara, CA, USA). Libraries exhibited visible size patterns and yields according to specifications (with major peaks expected around 142 bp for miRNAs and 152 for piwiRNAs). All libraries were indexed on the basis of barcoded adapter ligation. Libraries were combined to form independent pools. Next, an automated pooled library size selection was carried out. Each individual pool was size-selected for products between 115 and 165 bp with the Pippin Prep automated gel system and 3% agarose dye-free gel cassettes with internal standards (Sage Science, Beverley, MA, USA). Library pools were cleaned and concentrated by performing column centrifugation, prior to and after size selection, with the QIAquick PCR purification kit (Qiagen, Hilden, Germany). The size-selected library pool quality was reassessed with the Bioanalyzer High Sensitivity DNA Kit (Agilent Technologies). The final concentration was derived with qPCR, performed with the KAPA library quantification Kit for Illumina (Kapa Biosystems, Wilmington, MA, USA) on a 7900-HT PCR system (Applied Biosystems, Waltham, MA, USA, Thermo Fisher, Waltham, MA, USA). For each of the two pools, sequencing was performed on three independent HiSeq-2500 lanes (Illumina). Sequencing was performed on cBot and HiSeq-2500 instruments (Illumina), with the HiSeq single-read cluster generation, version 4, designed for the cBot-HiSeq, and the HiSeq sequencing-by-synthesis 50-cycle kit, according to manufacturer protocols. Per-sample raw-data files were generated in fastq format with bcl2fastq conversion software v1.8.4 (Illumina).

### 2.4. miRNA Sequence Analysis

All the analyses described here were performed with the previously described XICRA pipeline. Briefly, we checked the raw-data quality before and after trimming the reads to discard sequencing adapters using FASTQC (http://www.bioinformatics.babraham.ac.uk/projects/fastqc/, accessed on 10 October 2021) and cutadapt, respectively. We used miraligner, within the SeqBuster software v1, and the reference database miRBase (v22) for read mapping and assignment of miRNA identity (canonical and isomiR variants). Then, using miRTop, we formatted results into standardized files and generated expression count tables. Finally, for the analysis of small RNA DE, we employed DESeq2 provided in R software 4.2.3 (http://www.r-project.org, accessed on 30 May 2022). This differential analysis was adjusted for the following covariates: gender, age, and hemolysis. Statistical significance was determined by computing the adjusted *p*-value by the Benjamini and Hochberg method, which controls for the false discovery rate, to account for multiple tests. DE results were visualized using volcano and heat map plots. Volcano plots were generated using the EnhancedVolcano R package 4.2.3 (https://github.com/kevinblighe/EnhancedVolcano, accessed on 30 May 2022).

### 2.5. Statistical Analysis

Mean clinical parameter values were compared between groups with the Mann–Whitney test. Frequencies were compared between groups with the Chi-square test. ROC curves were performed with the Wilson–Brown method, with a 95% confidence interval (CI). All statistical analyses were performed with GraphPad Prism version 8.3.0 for Windows (GraphPad Software, San Diego, CA, USA).

## 3. Results

The plasma clinical and biochemical characteristics of the study patients and healthy controls are shown in Table 1. Compared with uninfected healthy donors, SARS-CoV-2-infected patients displayed significantly higher alanine aminotransferase (ALT) liver enzyme levels and lymphocyte and platelet counts (Table 1).

For the initial identification of miRNAs associated with SARS-CoV-2 infection, differential miRNA levels were determined by comparing plasma miRNA expression profiles between SARS-CoV-2-infected patients and the uninfected healthy control group. Because we aimed to use an unbiased systematic approach, large-scale deep-sequencing analysis of small RNA expression was performed on all individual plasma samples. With a mean of 4 million miRNA reads per sample, we identified a total of 1218 mature miRNAs in the study samples (median, 732 miRNAs; interquartile range [IQR]: 600–815). Cutoffs of effect size (fold change [FC] >1.2) and significance level (adjusted p [padj] < 0.05) were used to identify possible biomarker candidates. A set of 190 miRNAs were differentially expressed (DE) in COVID-19 patients, of which 83 were upregulated and 107 were downregulated (Figure 1 and Appendix A). These 190 deregulated miRNAs were DE expressed up to 304-fold (Appendix A). A clustered expression heat map, comparing expressions of the top-50 DE miRNAs, is shown in Figure 2A. The most highly upregulated miRNAs in COVID-19 patients were miR-122-5p, let-7b-5p, miR-146a-5p, miR-146b-5p, miR-629-5p, miR-24-3p, miR-12136, and let-7a-5p, while the most downregulated were miR-342-3p and miR-191-5p. These top-10 DE miRNAs had FC and padj values ranging from 153 to 5 and 2.51 × 10^−32^ to 2.21 × 10^−21^, respectively (Appendix A). To assess the potential of these 10 miRNAs as biomarkers of COVID-19 infection, receiver operating characteristic (ROC) curves were calculated. A ROC analysis demonstrated that miR-122-5p, let-7b-5p, miR-146a-5p, miR-146b-5p, miR-629-5p, miR-24-3p, miR-12136, let-7a-5p, miR-342-3p, and miR-191-5p levels individually discriminated hospitalized COVID-19 patients from healthy uninfected controls with high efficiency. These 10 miRNAs displayed areas under the curve (AUC) of 1.00 (95% CI 1.00–1.00), 1.00 (95% CI 1.00–1.00), 1.00 (95% CI 1.00–1.00), 1.00 (95% CI 1.00–1.00), 1.00 (95% CI 1.00–1.00), 0.97 (95% CI 0.93–1.00), 0.97 (95% CI 0.97–1.00), 0.98 (95% CI 0.96–1.00), 0.99 (95% CI 0.98–1.00), and 0.99 (95% CI 0.98–1.00), respectively, which unambiguously diagnosed the presence of SARS-CoV-2 infection (Appendix A).

The 40 studied COVID-19 patients were selected on the basis of their need for oxygenation/ventilation at the time of hospitalization. Thus, patients were categorized as hospitalized not requiring oxygen therapy (n = 6), hospitalized requiring low-flow oxygen (n = 23), and hospitalized requiring high-flow oxygen support (n = 11). The use of noninvasive oxygenation/ventilation strategies in the context of SARS-CoV-2 infection has been associated with better clinical outcomes in COVID-19 populations [16]. Patients requiring high-flow oxygen support differed significantly from those not requiring oxygen support or requiring low-flow oxygen in their baseline and 48 h oxygen saturation and arterial blood gas parameters (Table 2). Although our study patients differed in their clinical oxygen limits, oxygen saturation, fraction of inspired oxygen, and arterial blood gas, no other clinical or biochemical characteristics, including interleukin 6, ferritin, and D-dimer, distinguished those patients (Table 3). Importantly, these three parameters have been found to be significantly increased in severely ill patients or nonsurvivors [17]. This absence of disease markers led us to search for additional biomarkers. Thus, we next explored whether there was DE of plasma miRNAs between our hospitalized patients requiring oxygen support. Adjusting, again, for FC > 1.2 and padj < 0.05, 31 miRNAs were DE between patients requiring high or low oxygen (Figure 2B, Appendix A, Appendix A); 28 miRNAs were significantly upregulated and only 3 were downregulated. Not surprisingly, 15 of these 31 deregulated miRNAs were also DE between COVID-19 patients and the uninfected controls (Appendix A). A ROC analysis of the top-5 of the 31 DE miRNAs—miR-320d, miR-100-5p, miR-12136, miR-574-5p and miR-671-5p—revealed AUC values of 0.75 (95% CI 0.58–0.93), 0.70 (95% CI 0.52–0.88), 0.67 (95% CI 0.47–0.86), 0.71 (95% CI 0.52–0.90), and 0.67 (95% CI 0.49–0.85), respectively (Appendix A). Similarly, the comparison of miRNA plasma levels between patients requiring high-flow oxygen and those requiring no oxygen support showed the presence of six DE miRNAs (Figure 2C, Appendix A, and Appendix A). Two of these miRNAs, miR-100-5p and miR-671-5p, were also DE between patients requiring high or low oxygen support. Overall, these results demonstrated significant deregulation of circulating miRNAs in patients hospitalized for COVID-19 who required oxygen support.

## 4. Discussion

We identified 190 miRNAs that were robustly DE expressed up to 304-fold in our study cohort, demonstrating the diagnostic power of this miRNA signature. These findings extend previous reports confirming that circulating plasma miRNAs are valuable biomarkers of COVID-19 diagnosis [9,10,11,12]. In addition, we found a significant miRNA signature associated with patients requiring high-flow oxygen therapy after hospitalization, indicating that circulating miRNAs may be useful biomarkers of SARS-CoV-2 infection severity.

The leading deregulated plasma miRNAs identified here have been previously related to cancer and tissue inflammation. MiR-122-5p, let-7b-5p, let-7a-5p, miR-146b-5p, and miR-342-3p have been described as tumor suppressors and anti-inflammatory effectors [18]. Similarly, these five miRNAs have been found to display DE patterns in inflamed tissues. MiR-122 is the most abundant liver-tissue-specific miRNA. MiR-122-5p overexpression significantly restrains the proliferation, migration, and invasion of hepatocellular carcinoma (HCC) cells [18]. MiR-342-3p is also a potent tumor suppressor in HCC [18]. MiR-100-5p is associated with liver fibrosis progression and HCC [14,15,19]. Let-7b-5p and let-7a-5p act as tumor suppressors and are downregulated in many types of cancers. MiR-146a-5p, miR-629-5p, miR-24-3p, miR-191-5p, miR-320d, miR-574-5p, and miR-100-5p have also been related to cancer development. MiR-146a-5p promotes the initiation and progression of melanoma by activating Notch signaling [20]. Moreover, decreased interferon alpha inducible protein 27 levels were observed in COVID-19 patients with higher miR-146a-5p levels [21]. MiR-629-5p promotes the invasion of lung adenocarcinoma by increasing both tumor cell invasion and endothelial cell permeability [22]. MiR-24-3p expression is elevated in breast cancer patients with metastasis [23]. Similarly, miR-191-5p expression has been reported in various malignancies, such as breast, colon, lung, liver, prostate, pancreas, stomach, and ovarian cancers [24]. In most cases, the former miRNAs have been explored as biomarkers of tissue inflammation and cancer development. Interestingly, miR-12136 is related to the progression of intellectual disability, miR-574-5p is associated with schizophrenia, and miR-191-5p is implicated in cortical development, findings that suggest a relationship between SARS-CoV-2 infection and neurodegenerative disorders. Finally, miR-671-5p, which was significantly upregulated in the plasma of our hospitalized COVID-19 patients requiring high-flow oxygen support, is related to the progression of pulmonary inflammatory diseases [25]. Extracellular vesicles containing miR-671-5p alleviate lung inflammation and injury by regulating the AAK1/NF-κB axis [25]. In summary, the current data on the miRNAs described in our study indicate that SARS-CoV-2 infection impacts various tissues and pathogenic processes. At least 10% of SARS-CoV-2-infected individuals do not fully return to their baseline health after an acute COVID-19 illness [2]. These individuals show long-lasting symptoms for more than 12 weeks and most of them suffer functional disability. The real causes of the persistent symptomatology remain unknown, and, consequently, there is an urge to understand the origin of these persistent symptoms and to identify noninvasive prognostic biomarkers. There have recently been a range of experimental reports indicating the production of miRNAs by the SARS-CoV-2 virus itself [26,27]. However, searching for specific matches to all these described sequences in our trimmed reads gave no hits.

Farr and colleagues found that a 3-miRNA signature (miR-423-5p, miR-23a-3p, and miR-195-5p) independently classified COVID-19 cases with an accuracy of 99.9% [10]. MiR-423-5p and miR-23a-3p were also deregulated in our COVID-19 patients but were not on the top list. MiR-369-3p expression was altered in patients requiring mechanical ventilation in comparison with COVID-19 patients without this requirement [28], and miR-369-3p was also deregulated in our patients but, again, is not one of the most highly deregulated miRNAs in our study. A combination of high serum miR-22-3p and miR-21-5p and low miR-224-5p and miR-155-5p discriminated severe from mild/moderate COVID-19 [29]. MiR-22-3p and miR-224-5p were also deregulated in our study patients, but miR-21-5p and miR-155-5p were not. MiR-320d, which was highly upregulated in our patients requiring high-flow oxygen support, has been also found to be upregulated when patients with severe COVID-19 were compared with patients with mild COVID-19 [30]. Similarly, miR-3168 is upregulated in COVID-19 patients with pneumonia in comparison with healthy controls [31], and it was also upregulated in the patients described here. Finally, miR-122-5p and miR-12136, among the 10 top deregulated miRNAs in our patients with COVID-19, have recently also been associated with SARS-CoV-2 infection [32]. Although the miRNA signature associated with COVID-19 described in the former study is different from that described here, miR-370-3p, miR-885-5p, miR-193a-5p, and miR-483-5p, in addition to miR-122-5p and miR-12136, are also deregulated in both studies. A possible explanation for these discrepancies is that miRNA levels and types of expression, biogenesis, and sequestration change during the course of a coronavirus infection. Similarly, different virus variants may impact COVID-19 pathogenesis and, thus, circulating miRNA levels. For instance, it has been recently reported that patients’ risk of persistent COVID-19 symptoms 3 months after infection dropped from 46% with the original coronavirus strain and another called Alpha, to 35% with the Delta variant, to 14% with Omicron [33]. Additionally, a possible limitation of our study is that the healthy control group does not age-match the COVID-19 patient group. Similarly, our results may be also limited by the cross-sectional structure of our study. Further work should therefore evaluate the impact of different virus variants and vaccination and immunologic status on miRNA biogenesis, in order to use miRNAs as diagnostic tools for COVID-19 diagnosis and prognosis.

## Figures and Tables

**Figure 1 microorganisms-12-00440-f001:**
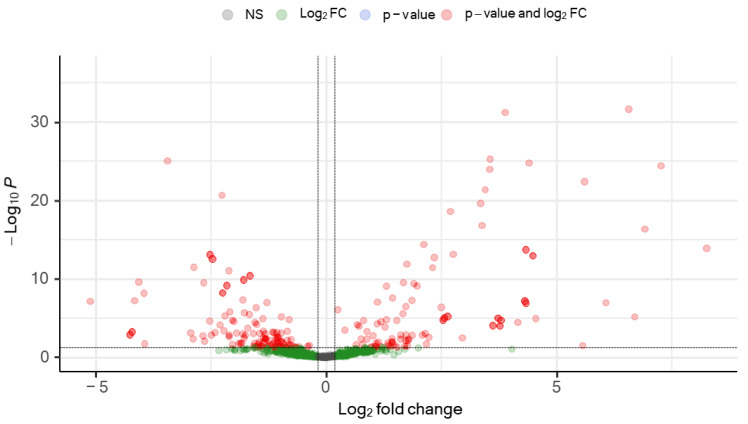
Volcano plot of differentially regulated baseline plasma circulating microRNAs in patients hospitalized with acute SARS-CoV-2 infection. Differential miRNA levels were determined by comparing plasma miRNA expression profiles between SARS-CoV-2-infected patients and the uninfected healthy control group. To identify biomarker candidates, cutoffs of effect size (fold change [FC] >1.2) and significance level (adjusted p [padj] < 0.05) were used. A set of 190 differentially expressed (DE) (red) miRNAs was identified.

**Figure 2 microorganisms-12-00440-f002:**
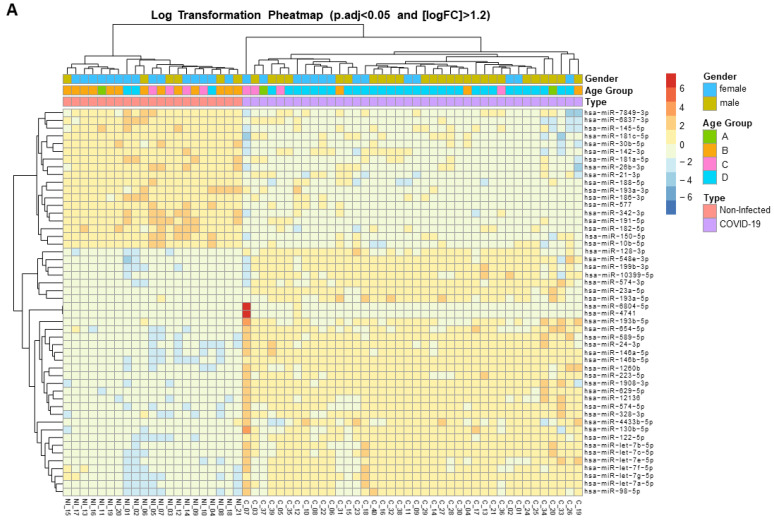
Expression heat map of deregulated miRNAs in the plasma of hospitalized patients with SARS-CoV-2 acute infection. (**A**) Heat map of the top-50 differentially expressed (DE) miRNAs from SARS-CoV-2-infected patients. (**B**) Heat map of the 31 DE miRNAs between patients with SARS-CoV-2 who required high-flow or low-flow oxygen support. (**C**) Heat map of the 6 DE miRNAs between SARS-CoV-2 patients requiring high-flow oxygen or no oxygen support. Heat maps show the miRNAs with significantly (fold change [FC] > 1.2 and adjusted by gender and age p [padj] < 0.05) different levels. Red: upregulated miRNAs; blue: downregulated miRNAs; yellow: no change.

**Table 1 microorganisms-12-00440-t001:** Clinical and biochemical characteristics of the whole study population according to SARS-CoV-2 infection status.

	SARS-CoV-2 Infected Patients	Uninfected Individuals	*p*-Value
N	40	21	-
Age yr, median (IQR) ^a^	56 (50.25–64)	37 (35–44.5)	<0.0001
Female no. (%) ^b^	14 (35)	15 (71.4)	0.0068
Albumin (g/L), median (IQR) ^a^	34.95 (31.10–37.78)	-	-
Bilirubin (mg/dL), median (IQR) ^a^	0.54 (0.42–0.64)	0.47 (0.37–0.74)	0.5944
Creatinine mg/dL, median (IQR) ^a^	0.75 (0.66–0.89)	0.78 (0.68–0.90)	0.7790
D-Dimer (µg/L), median (IQR) ^a^	453 (217–809)	-	-
Ferritin (ng/mL), median (IQR) ^a^	671 (315.1–1035)	-	-
Fibrinogen (mg/L), median (IQR) ^a^	654 (540–726)	-	-
Glucose (mg/dL), median (IQR) ^a^	102.20 (90.33–115)	-	-
ALT (U/L), median (IQR) ^a^	32 (19.75–58.50)	15 (12–19.50)	<0.0001
Potassium (mmol/L), median (IQR) ^a^	4.00 (3.71–4.36)	-	-
Sodium (mmol/L), median (IQR) ^a^	139.10 (136.80–141)	-	-
Urea (mg/dL), median (IQR) ^a^	32 (25–49)	-	-
Hematocrit %, median (IQR) ^a^	39.60 (34.60–44.90)	41.25 (38.40–43.63)	0.4708
Hemoglobin (g/dL), median (IQR) ^a^	13.40 (11.93–15.48)	14.20 (13.05–14.53)	0.4353
Mean Corpuscular Hemoglobin Concentration (g/dL), median (IQR) ^a^	30.20 (29.25–31.35)	30.35 (29.65–31.93)	0.4204
Leucocyte count (×10^9^/L), median (IQR) ^a^	5.50 (4.12–7.45)	6.50 (5.70–8.92)	0.0752
Lymphocyte count (×10^9^/L), median (IQR) ^a^	1.25(0.87–1.60)	2.00 (1.77–2.67)	<0.0001
Monocyte count (×10^9^/L), median (IQR) ^a^	0.45 (0.30–0.70)	0.50 (0.37–0.80)	0.3963
Platelet Distribution Width %, median (IQR) ^a^	17.20 (16.63–17.60)	16.10 (15.78–16.55)	<0.0001
Platelet (×10^9^/L), median (IQR) ^a^	214 (169.30–301.50)	253.50 (195.30–347.80)	0.1798
Interleukin 6 (pg/mL), median (IQR) ^a^	20.06 (9.53–52.69)	-	-

^a^ Mann–Whitney *t* test, ^b^ Chi-square test. interquartile range (IQR), alanine aminotransferase (ALT).

**Table 2 microorganisms-12-00440-t002:** Clinical oxygen characteristics of SARS-CoV-2-infected individuals receiving high-flow oxygen, low-flow oxygen, and no oxygen.

	High Flow	Low Flow	No Oxygen	*p*-Value ^a^	*p*-Value ^b^	*p*-Value ^c^
N (%)	11 (27.50)	23 (57.5)	6 (15)	-	-	-
Age, yr, median (IQR) ^d^	56 (44–65)	58 (51–64)	52 (35–58)	0.6174	0.362	0.0866
Female sex (%) ^e^	4 (36.40)	8 (34.8)	2 (33.33)	0.9281	0.9006	0.9470
Baseline SaO_2_ (%), median (IQR) ^d^	94 (91–96)	94(93–96)	96 (95.5–97)	0.4683	0.0376	0.0991
Baseline F_I_O_2_, median (IQR) ^d^	0.21 (0.21–0.21)	0.21 (0.21–0.21)	0.21 (0.21–0.21)	0.0980	0.5147	>0.9999
Baseline ABG PaO_2_ (mmHg), median (IQR) ^d^	52 (49–69)	69.5 (62.5–76.25)	76 (71–86)	0.0807	0.0353	0.1247
Baseline ABG F_I_O_2_, median (IQR) ^d^	0.21 (0.21–0.24)	0.21 (0.21–0.21)	0.21 (0.21–0.21)	0.0891	0.5089	>0.9999
Baseline ABG PaO_2_/F_I_O_2_, median (IQR) ^d^	246 (158–328.6)	317 (283.8–363.1)	361.9 (338–409)	0.0365	0.038	0.1091
48 h SaO_2_ (%), median (IQR) ^d^	95 (92–96)	96 (95–97)	96 (93.5–97.75)	0.0693	0.3848	0.6678
48 h F_I_O_2_, median (IQR) ^d^	0.4 (0.28–0.5)	0.28 (0.275–0.32)	0.21 (0.21–0.2275)	0.0171	0.0006	0.0058
48 h ABG PaO_2_ (mmHg), median (IQR) ^d^	75 (69–90)	79 (75–104)	74 (64–107)	0.3681	>0.9999	0.4982
48 h ABG F_I_O_2_, median (IQR) ^d^	0.4 (0.28–0.8)	0.28 (0.225–0.32)	0.21 (0.21–0.21)	0.0939	0.0165	0.0571
48 h ABG PaO_2_/F_I_O_2_, median (IQR) ^d^	194 (118–314)	304 (237–382.5)	352 (304–510)	0.0257	0.0385	0.2839
MDW (U), median (IQR) ^d^	22.24 (20.33–26.48)	23.76 (21–28.17)	24.57 (21.18–25.55)	0.3171	0.5035	0.7455
aPTT (s), median (IQR) ^d^	12.9 (11.7–14.15)	12.5 (11.8–13.2)	12.9 (12.28–14.08)	0.5675	0.7714	0.2284
aPTT (%), median (IQR) ^d^	78.5 (69–89.25)	82 (74–88)	77.5 (70.5–82.25)	0.6522	0.7717	0.1876
PT/INR, median (IQR) ^d^	1.21 (1.07–1.305)	1.15 (1.088–1.223)	1.29 (1.16–1.31)	0.5452	0.6273	0.1059
pH, median (IQR) ^d^	7.44 (7.435–7.47)	7.45 (7.4–7.475)	7.43 (7.365–7.465)	0.6684	0.3776	0.4402
Partial Pressure CO_2_ (mmHg), median (IQR) ^d^	43 (36.5–83)	38 (35.5–47.5)	35.5 (34.25–47.25)	0.3203	0.2567	0.4605
Partial Pressure O_2_ (mmHg), median (IQR) ^d^	73.5 (41.5–83.5)	69 (36–88)	90 (37.25–132.3)	0.7161	0.4356	0.4101
HCO_3_ (mmol/L), median (IQR)^d^	26.3 (23.5–29.38)	26 (24.05–27.75)	24.9 (22.55–27.7)	0.7313	0.4336	0.5992
Base excess (mmol/L), median (IQR) ^d^	3.2 (0.05–4.7)	2.1 (0.05–3.35)	0.6 (−1.375–2.275)	0.4220	0.1878	0.2553
O_2_ Saturation (%), median (IQR) ^d^	97.6 (96.18–99.7)	95.6 (78.2–98.3)	97.5 (35.4–98.2)	0.0302	0.5035	>0.9999

^a^ High flow vs. low flow, ^b^ high flow vs. no oxygen, ^c^ low flow vs. no oxygen, ^d^ Mann–Whitney *t* test, ^e^ Chi-square test. Interquartil range (IQR). Arterial blood gas (ABG), monocyte distribution width (MDW), activated partial thromboplastin time (aPTT), prothrombin time/international normalized Ratio (PT/INR).

**Table 3 microorganisms-12-00440-t003:** Clinical and biochemical characteristics of SARS-CoV-2-infected individuals receiving high-flow oxygen, low-flow oxygen, and no oxygen.

	High Flow	Low Flow	No Oxygen	*p*-Value ^a^	*p*-Value ^b^	*p*-Value ^c^
N (%)	11 (27.50)	23 (57.50)	6 (15)	-	-	-
Age yr, median (IQR) ^d^	56 (44–65)	58 (51–64)	52 (35–58)	0.6174	0.3620	0.0866
Female sex no. (%) ^e^	4 (36.40)	8 (34.80)	2 (33.33)	0.9281	0.9006	0.9470
Albumin (g/L), median (IQR)^d^	35.9 (25.75–42.85)	33.8 (30.23–36.2)	36.40 (32.90–36.90)	0.6778	>0.9999	0.3728
Bilirubin (mg/dL), median (IQR) ^d^	0.63 (0.54–0.7)	0.54 (0.41–0.61)	0.46 (0.35–0.55)	0.1058	0.0513	0.3789
Creatinine mg/dL, median (IQR) ^d^	0.71 (0.66–0.88)	0.75 (0.66–1.10)	0.77 (0.57–0.92)	0.5693	0.9578	0.8031
D-Dimer (µg/L), median (IQR) ^d^	704 (365–1546)	447 (266.5–764.8)	386.50(177.50–1415)	0.2513	0.5556	0.8272
Ferritin (ng/mL), median (IQR) ^d^	501 (375–901)	654 (309–994)	1005 (445.80–2025)	0.9543	0.3290	0.2300
Fibrinogen (mg/L), median (IQR) ^d^	619 (487–699)	598 (480–726)	685 (585.30–787.80)	0.7209	0.2635	0.2361
Glucose (mg/dL), median (IQR) ^d^	102.60 (88.45–131.60)	102.50 (91–120.30)	96.90 (82.93–142.30)	>0.9999	0.4559	0.4831
ALT (U/L), median (IQR) ^d^	30 (20.25–57.50)	32 (15.78–69.50)	45 (25.50–68.03)	0.9741	0.5440	0.5618
Potassium (mmol/L), median (IQR) ^d^	3.99 (3.79–4.28)	4.09 (3.71–4.43)	3.76 (3.53–4.12)	0.7265	0.2198	0.1018
Sodium (mmol/L), median (IQR) ^d^	138.10 (136.20–141.20)	139.50 (136.70–141.20)	138.70 (136.50–139.20)	0.7921	0.7472	0.2595
Urea (mg/dL), median (IQR) ^d^	33.50 (30.50–49.75)	31.95 (24.75–52)	26.10 (19.50–33.45)	0.6372	0.1274	0.1606
Hematocrit %, median (IQR) ^d^	44.90 (33–45.90)	38.70 (34.50–43.50)	39.75 (36.63–43.08)	0.3876	0.5740	0.6489
Hemoglobin (g/dL), median (IQR) ^d^	15.40 (11.10–15.90)	12.90 (11.90–14.50)	13.60 (12.25–14.95)	0.4397	0.6416	0.4855
Mean Corpuscular Hemoglobin Concentration (g/dL), median (IQR) ^d^	34 (33.60–34.70)	34.10 (33.40–34.50)	34.25 (33.73–34.65)	0.6303	0.7893	0.5190
Leucocyte count (×10^9^/L), median (IQR) ^d^	6.90 (4.10–8.70)	5.40 (4.20–7.50)	4.75 (3.67–6.95)	0.4183	0.2657	0.4214
Lymphocyte count (×10^9^/L), median (IQR) ^d^	1.30 (0.85–1.62)	1.25 (0.85–1.67)	1.35 (0.72–1.77)	>0.9999	0.9800	0.9239
Monocyte count (×10^9^/L), median (IQR) ^d^	0.70 (0.37–0.80)	0.45 (0.27–0.60)	0.40 (0.27–0.62)	0.0803	0.1485	0.8995
Platelet Distribution Width %, median (IQR) ^d^	17.40 (17.10–17.60)	17.20 (16.60–17.60)	16.90 (16.23–18)	0.2234	0.4461	0.9446
Platelet (×10^9^/L), median (IQR) ^d^	261 (145–316)	214 (168.30–303)	210.50 (175.30–222.30)	0.7108	0.3676	0.5003
Interleukin 6 (pg/mL), median (IQR) ^d^	26.60 (12.03–63.91)	19.10 (6.98–48.90)	19.06 (15.07–46.95)	0.3725	>0.9999	0.5808

^a^ High flow vs. low flow, ^b^ high flow vs. no oxygen, ^c^ low flow vs. no oxygen, ^d^ Mann–Whitney *t* test, ^e^ Chi-square test. interquartile range (IQR), alanine aminotransferase (ALT).

## Data Availability

MiRNA sequence data are publicly available in the GEO database under accession number GSE230092 (http://www.ncbi.nlm.-nih.gov/geo/, accessed on 19 April 2023).

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
