# Peer review of "Altered Plasma microRNA Signature in Hospitalized COVID-19 Patients Requiring Oxygen Support"

_microorganisms, 2024, doi:10.3390/microorganisms12030440_

Round 1

Reviewer 1 Report

Comments and Suggestions for Authors

This study of human encoded microRNAs during covid infection, studies a good number of patients is well presented and discussed and uses appropriate methodology.

I have one major concern which is that the comparison of the results is with a healthy non-infected control group. Therefore it is difficult to ascertain to what extent any changes in microRNAs are due to the acute illness and consequences of hospitalisation as opposed to changes specific to covid infection. To establish this would require comparison with a control group of hospitalised patients with a comparable severity of illness but not caused by covid-19. At least I think this limitation should be discussed in much greater detail though ideally analysis in such a control group should be presented. Therefore I do not think without such a control group it is possible to make the statement "Our large‐scale deep sequencing analysis identified a circulating miRNA signature 212 in all study patients that unequivocally and unambiguously diagnosed the presence of 213 SARS‐CoV‐2 infection"

The focus has been on human encoded microRNAs, There have recently been a range of reports indicating the production of microRNAs by the SARS-CoV-2 virus itself. It would be interesting to examine the small RNA-seq data for the presence and expression of SARS-CoV-2 encoded microRNAs.

It is difficult from the data presented to fully appreciate the AUC analysis of specific microRNAs and I would welcome a greater description and graphical presentation, perhaps a s a supplementary figure).

Some of the fold changes in expression are very high. It would be helpful to understand the sequence counts in each individual for such microRNAs to better understand the extent to which there is no or low level detection of these sequences in individual patients.

Independent validation of specific highly regulated microRNAs with a different technique and in a replication cohort would provide much greater confidence in the results

Reviewer 2 Report

Comments and Suggestions for Authors

The authors have performed a well-structured screening for identifying the miRNA signature in SARS-CoV-2 infected individuals. Additionally, authors have performed a classification of SARS-CoV-2 infected individuals into three categories based on their need for oxygenation at the time of hospitalization. Based on this classification, they have identified differentially expressed miRNAs between patients who needed high oxygen vs patients who did not need oxygen support, and between patients who needed high oxygen compared to those needing low oxygen during their hospitalization.

The identification of these miRNAs along with similar studies performed previously, as cited in the manuscript, provide rich foundation for future research to analyze the functional role of these miRNAs in the SARS-CoV-2 disease pathogenesis.

With that said, I have a few comments that can be looked at by the authors to refine the manuscript:

Comments:

1.       The median age of infected patients in the study is 56, while that of uninfected is 37 which is a huge primary difference between the test and control dataset. It will be appreciated if the authors can comment (in discussion) on how this difference may have led to a bias on the results and can suggest for future studies on more comparable age-group datasets.

2.       It will be helpful to readers, if the authors can provide an index for all terms (parameters) used in Table 2 for evaluation of clinical oxygen, or just mention them at the bottom of this table as done for ALT in Table 1.

3.       After how many days of hospitalization, were the plasma samples collected for each patient. It will be good to mention this aspect in the study cohort section of methods.

4.       Referring back to previous comment, the authors should discuss about the possibility of change in miRNA profile from start of infection to a course of time. As the authors have studied the profile of patients only once during the time of infection, it is yet to be understood that for how long does this miRNA signature persist after the infection.

5.       I would like to recommend that authors write more (may need to find more literature) about the expression of identified miRNAs during other viral infection to provide more corresponding literature of these miRNAs during immune response. These citations if added, can replace the ones that describe the tumor-suppressive or oncogenic nature of respective miRNAs, which I find as less relevant with regards to the study.

Round 2

Reviewer 1 Report

Comments and Suggestions for Authors

I am concerned that the control group is inappropriate to define whether the microrna changes are associated with covid-19 infection (as opposed to any acute illness). Without that control, the statements such as "the miRNAs described here and elsewhere may be an effective tool for the prognosis and diagnosis of COVID‐19 severity and persistent symptoms" and "Our large‐scale deep sequencing analysis identified a circulating miRNA signature in all study patients that can diagnose the presence of SARS‐CoV‐2 infection."  are not warranted.

Author Response

As indicated by this reviewer the statements "the miRNAs described here and elsewhere may be an effective tool for the prognosis and diagnosis of COVID‐19 severity and persistent symptoms" and "Our large‐scale deep sequencing analysis identified a circulating miRNA signature in all study patients that can diagnose the presence of SARS‐CoV‐2 infection." have been now deleted from the discussion section, line 261 and line 218, of the new version, respectively.